**PLOS** NEGLECTED TROPICAL DISEASES

# Transmission ecology of *Trypanosoma cruzi* by *Rhodnius prolixus* (Reduviidae: Triatominae) infesting palm-tree species in the Colombian Orinoco, indicates risks to human populations

Plutarco Urbano[1,2,3], Carolina Hernández[3,4,5], Natalia Velásquez-Ortiz[3], Nathalia Ballesteros[3], Luisa Páez-Triana[3], Laura Vega[3], Vanessa Urrea[3], Angie Ramírez[3], Marina Muñoz[3], Carlos N. Ibarra-Cerdeña[6], Camila González[1], Juan David Ramírez [3,4]*

**1** Centro de Investigaciones en Microbiología y Parasitología Tropical (CIMPAT), Departamento de Ciencias Biológicas, Universidad de los Andes, Bogotá, Colombia, **2** Grupo de Investigaciones Biológicas de la Orinoquia, Universidad Internacional del Trópico Americano (Unitrópico), Yopal, Colombia, **3** Centro de Investigaciones en Microbiología y Biotecnología-UR (CIMBIUR), Facultad de Ciencias Naturales, Universidad del Rosario, Bogotá, Colombia, **4** Molecular Microbiology Laboratory, Department of Pathology, Molecular and Cell-Based Medicine, Icahn School of Medicine at Mount Sinai, New York City, New York, United States of America, **5** Centro de Tecnología en Salud (CETESA), Innovaseq SAS, Bogotá, Colombia, **6** Departamento de Ecología Humana, Centro de Investigación y de Estudios Avanzados del Instituto Politécnico Nacional (Cinvestav), Mérida, Yucatán, México

* juand.ramirez@urosario.edu.co, juan.ramirezgonzalez@mssm.edu

## Abstract

### Background

Chagas disease, affecting approximately eight million individuals in tropical regions, is primarily transmitted by vectors. *Rhodnius prolixus*, a triatomine vector, commonly inhabits in ecotopes with diverse palm tree species, creating optimal conditions for vector proliferation. This study aims to explore the transmission ecology of *Trypanosoma cruzi*, the causative parasite of Chagas disease, by investigating the feeding patterns and natural infection rates of *R. prolixus* specimens collected from various wild palm species in the Colombian Orinoco region.

### Materials and methods

To achieve this objective, we sampled 35 individuals from three palm species (*Attalea butyracea*, *Acrocomia aculeata*, and *Mauritia flexuosa*) in a riparian forest in the Casanare department of eastern Colombia, totaling 105 sampled palm trees. DNA was extracted and analyzed from 115 *R. prolixus* specimens at different developmental stages using quantitative PCR (qPCR) for *T. cruzi* detection and identification of discrete typing units. Feeding preferences were determined by sequencing the 12S rRNA gene amplicon through next-generation sequencing.

### Results

A total of 676 *R. prolixus* specimens were collected from the sampled palms. The study revealed variation in population densities and developmental stages of *R. prolixus* among

**Data Availability Statement:** All the data is available in the manuscript and the supplementary files.

**Funding:** We thank the Direccion de Investigacion e Innovación from Universidad del Rosario (JDR) for funding the study. We thank the CEIBA foundation for funding the PhD studies of PU. The funder had no role in the study design, data collection and analysis, decision to publish, or preparation of the manuscript.

**Competing interests:** The authors have declared that no competing interests exist.

palm tree species, with higher densities observed in *A. butyracea* and lower densities in *M. flexuosa*. TcI was the exclusive *T. cruzi* discrete typing unit (DTU) found, with infection frequency positively correlated with *R. prolixus* abundance. Insects captured in *A. butyracea* exhibited higher abundance and infection rates than those from other palm species. The feeding sources comprised 13 mammal species, showing no significant differences between palm species in terms of blood sources. However, *Didelphis marsupialis* and *Homo sapiens* were present in all examined *R. prolixus*, and *Dasypus novemcinctus* was found in 89.47% of the insects.

## Conclusion

This study highlights the significance of wild palms, particularly *A. butyracea*, as a substantial risk factor for *T. cruzi* transmission to humans in these environments. High population densities and infection rates of *R. prolixus* were observed in each examined palm tree species.

## Author summary

Previous studies on Chagas disease in the Orinoco region primarily focused on domestic and peri-domestic areas of rural residences. However, to fully grasp the dynamics of *T. cruzi* transmission from natural, preserved areas to humans, it is crucial to comprehend the habitats and food resources that kissing bugs encounter in their natural distribution zones. In this study, we delved into both the habitat conditions and the food sources available on wild palms in the eastern plains region of Colombia, with the aim of understanding the establishment of the Chagas disease vector. Notably, our findings revealed that the studied kissing bugs fed on both wild and domestic animals, as well as humans, across all stages of their development. Furthermore, this research identified differences in the structural physiognomy of wild palms directly linked to the establishment of bug colonies. This knowledge enhances our understanding of the role of wild habitats in the dynamics of *T. cruzi* transmission in endemic areas.

## Introduction

Chagas disease, a significant global health concern, affects approximately 8 million people worldwide, resulting in an annual mortality of 10,000 in Latin America, with 65 million individuals at risk of infection [1,2]. Over the past four decades, Colombia has reported a consistent prevalence of 2%, accompanied by a mortality rate of 0.211 individuals per 100,000 inhabitants [3]. Notably, the Orinoco region in eastern Colombia stands out as an endemic area with the highest Chagas disease mortality rate [3,4].

This disease is caused by the protozoan *Trypanosoma cruzi*, an intracellular parasite that infects various mammals, including humans [5]. The life cycle of *T. cruzi* encompasses various stages, starting from the infective form found in the feces of vectors to intracellular replication and transmission through the bloodstream [6]. Despite numerous described infection mechanisms, vector transmission by triatomine insects remains widespread in endemic regions such as the Orinoco [7,8]. Triatomine insects have undergone physiological and behavioral adaptations to survive and thrive in diverse environments, ensuring increased interaction with vectors and parasite reservoirs [9–11]. While, on a larger scale, vectors are associated with

different ecotopes and display a degree of tolerance for ecotope degradation [12], on a more detailed level, their presence hinges on specific ecological conditions that foster persistence and biological success [13–16].

Palm trees play a crucial role in shaping the distribution of triatomine insects throughout the Americas [17]. There are around 27 triatomine species belonging to five tribes that have been identified as colonizing palm trees, with a particular preference for the *Attalea* genus [13,17]. The importance of wild palms as natural habitats for sustaining triatomine colonies is highlighted by the favorable habitat and microclimatic conditions they offer [14–15,18–22]. This is attributed to the regular reproductive events and leaf abscission of palm trees, leading to the accumulation of biomass at the base of the crown [23–25].

The stability of microclimates provided by palms is essential for understanding triatomine population dynamics, distribution, and colonization, given their sensitivity to fluctuations in temperature and humidity [14,26,27]. Ecotopes formed by palms play a pivotal role in shaping the distribution of the *Rhodnius* genus [13]. The associations between triatomines and palms may be attributed to the structural and reproductive characteristics of the palms themselves [14,23,28,29]. An examination of palm distribution in the Casanare department of the eastern plains of Colombia reveals the presence of 26 wild palm species. Among them, *Mauritia flexuosa*, *Acrocomia aculeata*, and *A. butyracea* stand out in terms of density and distribution. These palm species are identified as sources of triatomines in this endemic area [13].

Wild palms function as habitats for a diverse array of vertebrates, serving as nesting sites or occasional hosts for *T. cruzi*. This includes reservoir species (mammals) and non-reservoir species (birds) [14,16,24,26,30,31]. The presence of vertebrates within palm crowns is influenced by changes in response to the structural physiognomy of palms [28,32], thereby maintaining conditions favorable for vector reproductive success and parasite transmission [14,31]. In the Colombian Orinoco and Brazilian Amazonia, various vertebrates have been identified as primary feeding sources for triatomines associated with palms [33–37]. The interaction between reservoirs and triatomines within palm crowns categorizes wild palms as a risk factor for parasite transmission from sylvatic environments to human dwellings [15,37–39]. This positions them as dispersion centers for infected triatomines carrying *T. cruzi* [32,39].

Extensive investigations in the Colombian Orinoco, with a specific focus on *R. prolixus* associated with *A. butyracea*, indicate that this palm serves as a key indicator of *T. cruzi* transmission risk [13,23,37,39–41]. *A. butyracea* creates optimal conditions for the biological success of *R. prolixus*, establishing itself as the predominant vector of *T. cruzi* in the plains of Colombia and Venezuela [13,39]. However, a comprehensive understanding of palm infestation, *T. cruzi* infection, transmission dynamics, and potential reservoirs requires further exploration, especially for palm species belonging to genera such as *Acrocomia*, *Mauritia*, *Euterpe*, *Oenocarpus*, and *Cocos*.

These palm species have been identified as risk factors for *T. cruzi* transmission in Colombia, Brazil, and Venezuela [20,24,26,34,39,42–45]. In this broader context, we hypothesize that each palm species presents a unique risk, with its structural physiognomy directly correlated with insect establishment, population densities, infection rates, and food supply. To test this hypothesis, our study aims to investigate the variation in the transmission ecology of *T. cruzi* among different wild palm species in the Colombian Orinoco, specifically characterizing feeding sources and natural infection of *R. prolixus* collected from *M. flexuosa*, *A. aculeata*, and *A. butyracea*.

## Materials and methods

### Ethics statement

The insects were collected from public land in Colombia through diverse entomological surveillance techniques. The authorization from ANLA (Autoridad Nacional de Licencias

Ambientales) 1177–2014 (IDB0359) was provided by UNIVERSIDAD DE LOS ANDES, allowing for the lawful and regulated undertaking of the collection activities.

## Study area

Sampling was conducted in Aguazul, a municipality situated in the Casanare department near the floodplains of the Cusiana River, at an average altitude of 190 meters. The landscape comprised natural and gallery forests exhibiting varying degrees of human impact. Aguazul experiences a rainy season spanning from April to October, with an average rainfall of 305 mm, followed by a dry season for the remaining months, characterized by an average rainfall of 68.9 mm. The annual temperature in the region is 27°C, and the monthly precipitation averages 204.9 mm [46].

Specifically, we selected a 43-hectare fragmented secondary forest located at coordinates N: 4.524129, W: -72.241668 (Fig 1), featuring the presence of *A. butyracea*, *A. aculeata*, and *M. flexuosa* palms. This forest is adjacent to a highly fragmented area designated for pasture and livestock establishments, which includes 17 human dwellings with documented previous instances of triatomine presence. Sampling activities were carried out in January 2020, chosen deliberately during the low rainfall period for enhanced accessibility to the study site. This timeframe also aligns with a period of increased triatomine abundance, as indicated by previous research [38].

## Triatomine sampling

We conducted a comprehensive sampling effort, targeting 35 palms of each species (*A. butyracea*, *A. aculeata*, and *M. flexuosa*), resulting in a total of 105 individuals. The capture of *R. prolixus* was accomplished through a live bait trap, following the methodology outlined by Angulo and Esteban [47]. Each trap was strategically placed in the crown of every palm, spanning a 12-hour period from 18:00 to 6:00 the following day.

Upon capture, individuals from each palm were categorized based on their developmental stage, including first instar nymph (N1), second instar nymph (N2), third instar nymph (N3), fourth instar nymph (N4), fifth instar nymph (N5), and adults. Subsequently, the collected specimens were meticulously preserved in Eppendorf tubes containing absolute ethanol, appropriately labeled, and then transported to the Microbiology and Biotechnology Research Center at Universidad del Rosario (CIMBIUR) for subsequent molecular analysis.

To evaluate the prevalence of *R. prolixus*, we computed four entomological indices: infestation (calculated as the number of positive palms divided by the total number of palms sampled, multiplied by 100), clustering (determined by the number of recollected individuals divided by the number of infested palms), colonization (expressed as the number of palms with nymph presence divided by the total number of positive palms, multiplied by 100), and density (calculated as the number of individuals collected divided by the total number of palms sampled). These indices were derived from the criteria established by Suarez-Dávalos et al. [48] and Urbano et al. [41]. A palm was deemed positive when at least one individual from any developmental stage was collected. Taxonomic identification was carried out using a stereoscope, adhering to morphological guidelines from the Lent and Wygodzinsky [49] taxonomic key.

## Microclimate and physiognomy of the palms

For each individual palm per species, we installed a Data Logger (iButton) in the crown to continuously record relative humidity and temperature at one-hour intervals throughout the sampling days (24 hours a day). As a control measure, an additional Data Logger was placed in the sampling area during the same period. Average values per hour for both relative humidity and

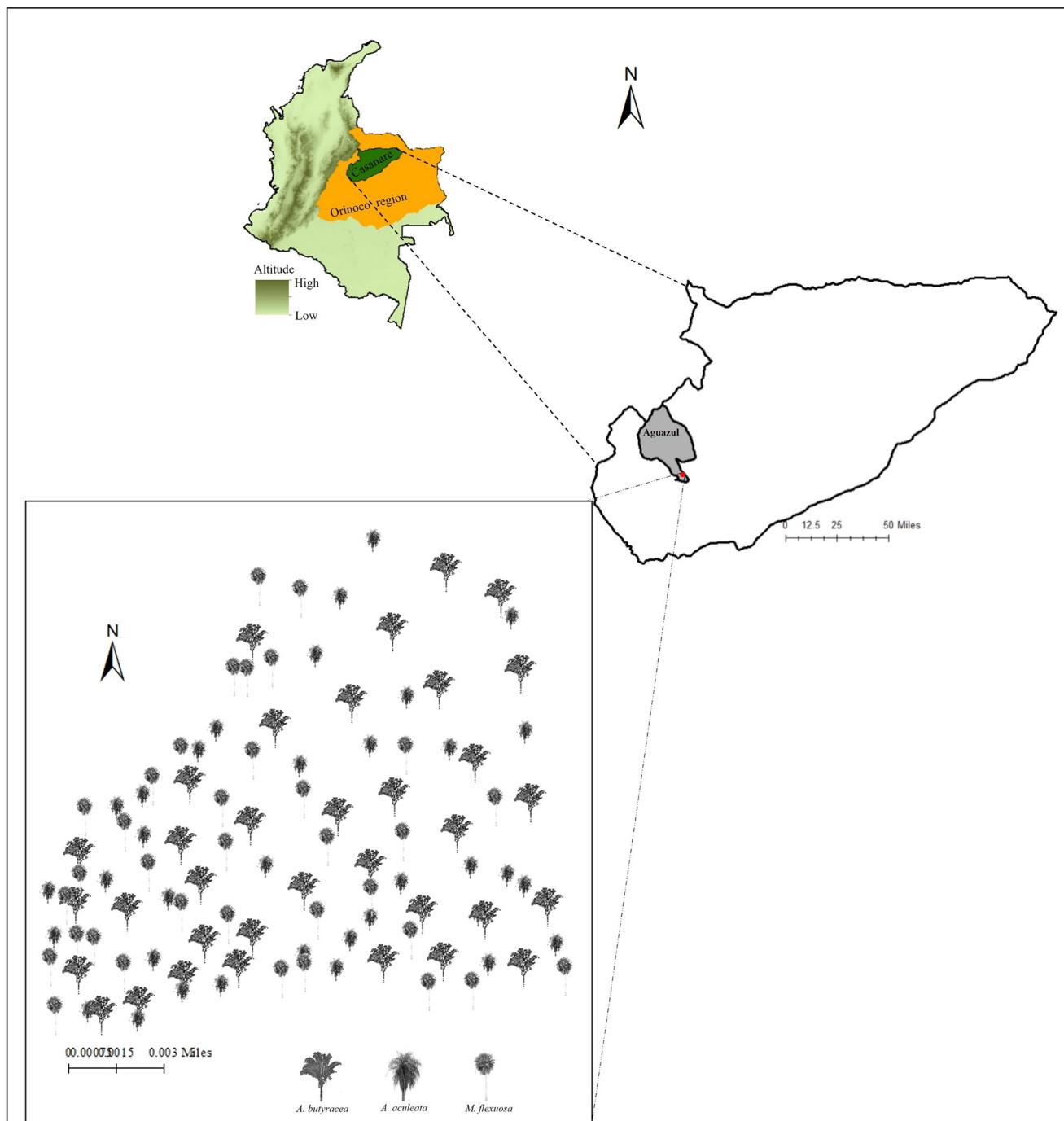

**Fig 1. Aguazul municipality map in Casanare department.** Shows the geographical location of the sample site. Red circle indicate the midpoint of the samples area. The Orinoco region is marked in orange on the map of Colombia. The map was constructed using QGIS version 2.18.7. Basemap: Elevation/World_Hillshade https://bit.ly/3vVQ1lL; Sources: Esri, Airbus DS, USGS, NGA, NASA, CGIAR, N Robinson, NCEAS, NLS, OS, NMA, Geodatastyrelsen, Rijkswaterstaat, GSA, Geoland, FEMA.

temperature were calculated for each palm species and the external environment [23,34]. This approach facilitated the comparison of variations in these variables recorded in the palms with those in the external forest environment. In addition to climatic data, we recorded physiognomic variables such as palm height, diameter at breast height (DBH), number of leaves, and palm top size for each sampled individual of the three palm species [23,50,51]. These recorded variables contribute to a comprehensive understanding of the ecological context and conditions associated with the sampled palms.

## Detection and genotyping of *T. cruzi*

The DNA extraction process involved obtaining genetic material from the abdominal section of 115 triatomines, distributed among different palm species (18 from *M. flexuosa*, 56 from *A. aculeata*, and 41 from *A. butyracea*). This extraction was carried out using automated procedures on the Hamilton Microlab STAR robot with The Quick-DNA/RNA MagBead kit from Zymo Research. Here, we included 5 extraction controls (RT-PCR Grade Water from Invitrogen was utilized, undergoing comprehensive testing for both prokaryotic and eukaryotic genomic DNA contamination, assessed through 16S rRNA and 18S rRNA evaluations) that were all negative suggesting very low levels of laboratory contamination. For the detection and quantification of satellite DNA of *T. cruzi*, we employed quantitative PCR (qPCR). Parasitic loads were measured as parasite equivalents per mL, following the established protocol by Velásquez-Ortiz et al. [52]. A TcI strain (MHOM/CO/04/MG) served as the standard curve for qPCR [52]. qPCR-positive samples underwent genotyping using conventional PCR to amplify the spliced leader intergenic region of the miniexon gene (SL-IR). The resulting PCR products were subjected to electrophoresis, and genotypes were verified by visualizing a 300 bp product for TcII and a 350 bp product for TcI [53]. In our study, positive controls included *T. cruzi* DNA obtained from culture, while negative extraction controls utilized DNA from *R. prolixus* specimens sourced from a colony free of *T. cruzi* infection. Furthermore, qPCR negative controls were executed using PCR Grade Water. It is essential to highlight that all results were deemed valid only if the positive and negative controls performed as expected, ensuring the reliability of the outcomes.

## Feeding sources molecular characterization

Positive DNA samples were categorized into 18 pools based on palm species, life stages (I-V instar nymphs and adults), and sex (S1 Table). The pooled DNA was subjected to sequencing, targeting a 215 bp fragment of the 12S rRNA gene, using Illumina NovaSeq 6000 (Novogene Co., Ltd). To ensure the reliability of our Amplicon-based Next Generation Sequencing for the 12S rRNA gene, we incorporated Genomic DNA-Rat Male Biochain [54] as the positive control. For negative control purposes, RT-PCR Grade Water from Invitrogen was utilized, undergoing comprehensive testing for both prokaryotic and eukaryotic genomic DNA contamination, assessed through 16S rRNA and 18S rRNA evaluations [55]. It is crucial to note that negative controls were void of amplicons or reads from human DNA. Additionally, DNA from the *R. prolixus* colony (only fed with *Gallus gallus* blood) was utilized to identify reads associated with *Gallus gallus*, confirming it as the feeding source for these insects.

Quality control of the sequencing products was diligently executed using FastQC and MultiQC. Following this, QIIME was employed to eliminate barcodes, and all sequences underwent BLASTn analysis, comparing them against a reference dataset consisting of 137 vertebrate sequences, as outlined by Arias-Giraldo et al. [33]. The estimation of vertebrates present in the triatomine diet was determined based on the number of reads per vertebrate, serving as a reliable proxy [56]. The DADA2 pipeline was instrumental in determining the

frequency of each amplicon sequence variant (ASV) in every sample. The resulting ASV table, including the number of reads, was transformed into relative values using RStudio, enabling the approximation of relative abundance per sample.

## Statistical analysis

To assess variations in population density and developmental stages across palm species, a series of statistical tests were performed. Normality was examined using the Shapiro-Wilk test, and homoscedasticity was evaluated through Levene tests. Population density differences based on the sampling palm were observed using Kruskal-Wallis tests. Multiple comparison tests, including post hoc Dunn analysis with Bonferroni correction, were conducted to identify any variations in the density of *R. prolixus* developmental stages per palm species. The same statistical tests were applied to compare parasite loads, considering both palm species and the age structure of *R. prolixus*. For the analysis of structural physiognomy variation by palm species, a non-parametric analysis was conducted, comparing palm height, diameter at breast height (DBH), number of leaves, and palm top size. This comparative analysis utilized data from the 35 individuals sampled for each species, following methodological parameters outlined by Urbano et al. [23]. Additionally, a Kruskal-Wallis (KW) analysis was performed to compare temperature and humidity behavior per day for each palm species and the external environment. This analysis used hourly averages recorded during the 24 hours of the sampling days. To explore similarities among palm species based on measured variables and the abundance of infected vs. non-infected R. *prolixus*, a linear discriminant analysis (LDA) was executed. Finally, RCircos [57] and ggplot packages [58] of R software version 4.2.2 were employed to visually represent the feeding sources found in *R. prolixus* specimens collected in each palm species. All statistical analyses were conducted with a 95% confidence level (p < 0.05).

## Results

### Microclimate and physiognomy of the palms

In our findings, *A. butyracea* exhibited significantly higher DBH values compared to the other species (Dunn, p < 0.0001) (Fig 2A). *Mauritia flexuosa* displayed significantly lower crown volume and a reduced number of leaves compared to the other two species (Dunn, p < 0.0001) (Fig 2B and 2D). *Acrocomia aculeata* demonstrated significantly lower heights than *M. flexuosa* and *A. butyracea* (Dunn, p < 0.0001) (Fig 2C). Despite variations in some physiognomic variables among the three palm species, the Linear Discriminant Analysis (LDA) did not indicate structural differentiation among the species.

Regarding environmental variables, *A. butyracea* and *A. aculeata* exhibited significantly higher temperatures than *M. flexuosa* and the control placed in surrounding areas (Kruskal-Wallis, p = 0.009; Dunn, p < 0.0001) (Fig 2E). Additionally, relative humidity was higher at night for *A. butyracea* (Kruskal-Wallis, p = 0.0061; Dunn, p < 0.001) (Fig 2F). The temperature of *A. butyracea* varied on average by 2.25˚C from day to night, 3.32˚C *in A. aculeata*, 4.24˚C in *M. flexuosa*, and 5.03˚C in the surrounding area. The same pattern held for relative humidity, indicating that *A. butyracea* exhibits higher microclimate stability (Fig 2E and 2F).

### Population density of *R. prolixus* per palm species

In total, 676 *R. prolixus* specimens were collected, consisting of 609 nymphs (90.10%) and 67 adults (9.9%). The distribution across palm species was as follows: 379 individuals (56.1%) in *A. butyracea*, 264 (39%) in *A. aculeata*, and 33 (4.9%) in *M. flexuosa*. Significant variations

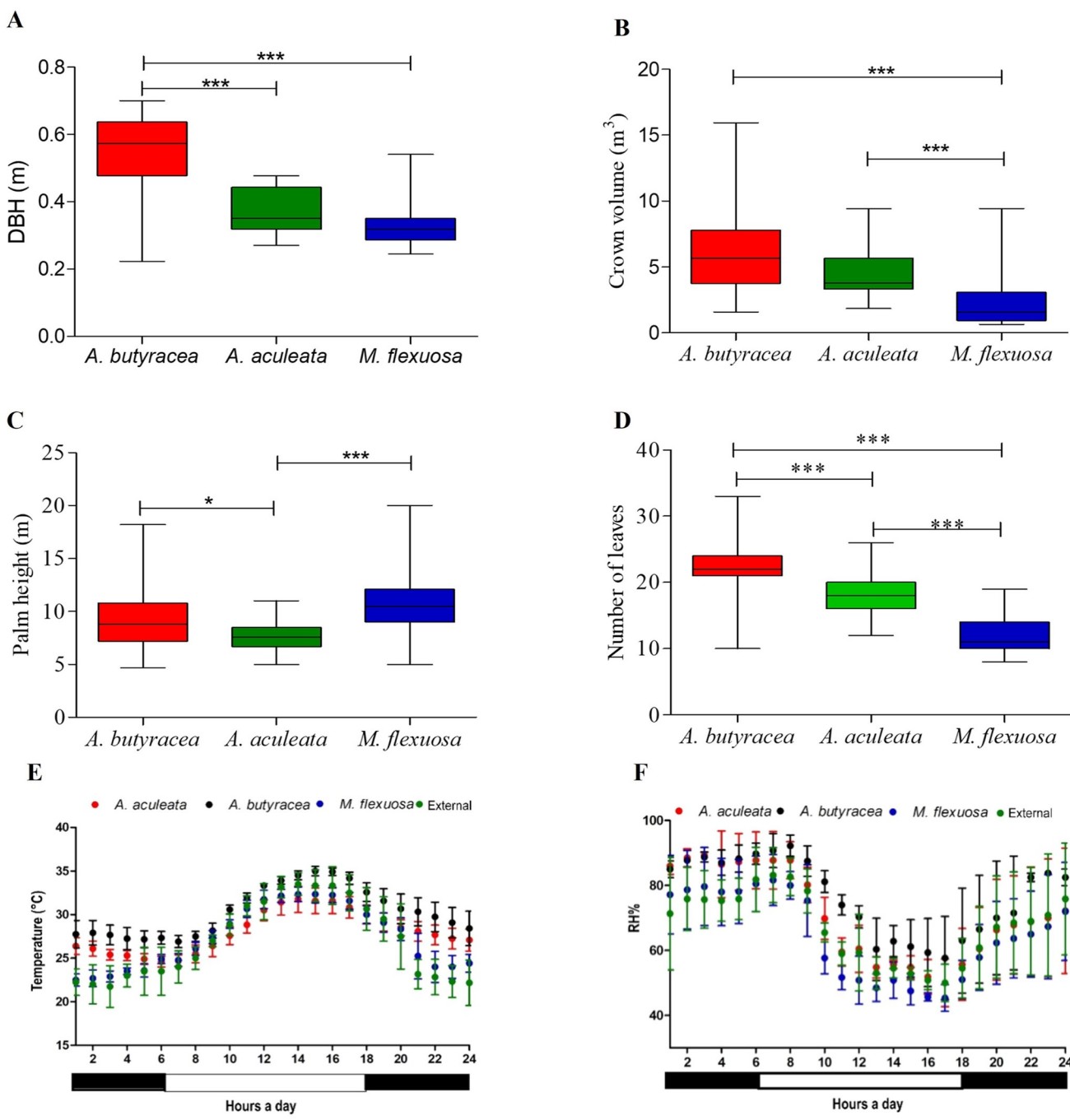

**Fig 2. Comparison of habitat conditions and microclimates of the studied palms.** (A-D) Phenotypic characteristics of the palm species, daily temperature (E) and humidity (F) for the three palm species and the external environment. Top and bottom lines represent the maximum and minimum values, respectively. The middle line in the boxes represent the median of each data groups. The asterisks (***) indicate the significative differences (Dunn test p<0.0001) for 35 palms per species.

were observed in the number of individuals per palm species (Kruskal-Wallis, p < 0.0001). Both *A. butyracea* and *A. aculeata* exhibited similar, higher numbers of insects compared to *M. flexuosa* (Dunn, p < 0.0001) (Fig 3A). Similar patterns were noted for developmental stages, with significantly lower densities in *M. flexuosa*, except for the N5 instar (Kruskal-

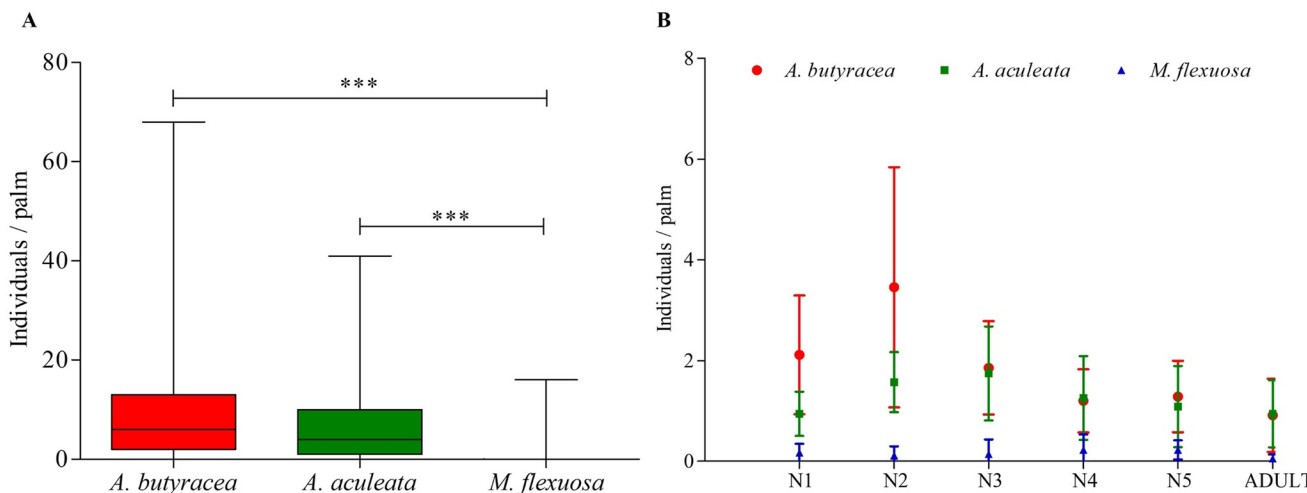

**Fig 3. Density of *R. prolixus* and its developmental stages according to the sampled palm.** (A) *R. prolixus* density according to the sampled palm. (B) *R. prolixus* developmental stages density according to the palm species. Top and bottom lines represent the maximum and minimum values, respectively. The middle line in the boxes represent the median of each data groups. The asterisks (***) indicate the significative differences (Dunn test p<0.0001) for 35 palms per species.

Wallis, p = 0.005; Dunn, p < 0.0001). Bugs collected in *A. butyracea* showed higher densities of N1 and N2 instars compared to *A. aculeata*, while the remaining developmental stages were similar in these two palm species (Fig 3B). In summary, the results indicate that *R. prolixus* exhibited higher densities in *A. butyracea* compared to the other palm species (S2 Table).

## Entomological indices of *R. prolixus*

The infestation index ranked highest in *A. butyracea*, followed by *A. aculeata* and *M. flexuosa*. Colonization index values were lower for *M. flexuosa* (87.5%) compared to the other species, which exhibited a 100% colonization rate. The clustering index showed the highest values for *A. butyracea*, followed by *A. aculeata* and *M. flexuosa*. Notably, the density index, representing the number of individuals per sampled palm, was markedly lower in *M. flexuosa*—11.5 times less than *A. butyracea* and 8.02 times less than *A. aculeata* (Table 1).

## Infection index and quantification of *T. cruzi*

We selected individuals for *T. cruzi* detection, ensuring representation from all stages of *R. prolixus* across various palm species. Our approach involved choosing a minimum of 10% and up to 100% of insects from each stage within every palm species. This selection process was based on estimating the sample size using the most recent *T. cruzi* positivity rate in *R. prolixus* individuals from palms in Casanare, which was 85.6%. With a 90% confidence level and an accepted error of 5.5%, we determined that a sample size of 115 individuals would be

**Table 1. Entomological indices and *T. cruzi* infection of triatomines collected in wild palms in the Colombian Orinoco.**

| Palm species | Vector species | Infestation (%) | Colonization (%) | Clustering (%) | Density (%) | *T. cruzi* infection |
|---|---|---|---|---|---|---|
| *A. butyracea* | *R. prolixus* | 94.29 | 100 | 11.48 | 10.82 | 100% (41/41) |
| *A. aculeata* | *R. prolixus* | 80 | 100 | 9.42 | 7.54 | 96.42% (54/56) |
| *M. flexuosa* | *R. prolixus* | 22.86 | 87.5 | 4.71 | 0.94 | 88.88% (16/18) |

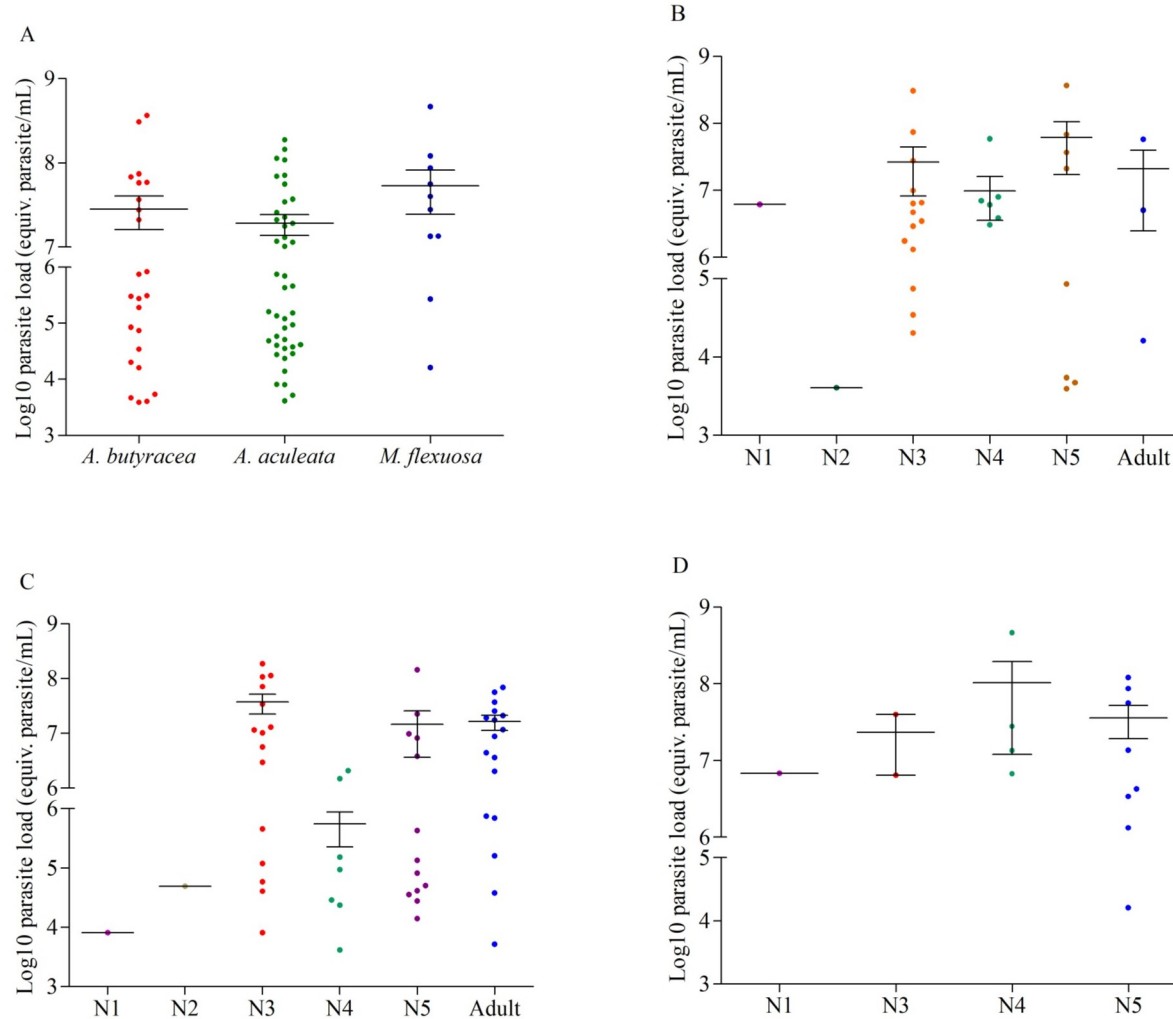

**Fig 4. Comparative analysis of the parasite loads of *R. prolixus*.** (A) Shows the behavior of the parasite loads of all the samples processed in the three species of palms. (B-D) Show the parasite load by developmental stage in (B) *A. butyracea*, (C) *A. aculeata* and (D) *M. flexuosa*.

sufficient. Specifically, the distribution of individuals was as follows: 54% (18/33) for *M. flexuosa*, 21.2% (56/264) for *A. aculeata*, and 11% (41/379) for *A. butyracea*.

In our analysis of 115 examined *R. prolixus* individuals, 96.52% (111/115) tested positive for *T. cruzi* infection. The infection rates by palm species were as follows: *A. butyracea* 100% (41/41 insects), *A. aculeata* 96.42% (54/56 insects), and *M. flexuosa* 88.88% (16/18). While the prevalence was higher in *R. prolixus* captured in *A. butyracea* (1; 95% CI: 0.91–1) than in *A. aculeata* (0.96; 95% CI: 0.89–0.99) and *M. flexuosa* (0.89; 95% CI: 0.67–0.97), no significant differences were found among them. All positive samples were identified as the TcI DTU. The comparative analysis of total parasite loads in *R. prolixus* captured in the three palm species did not show significant differences (Kruskal-Wallis; p = 0.07) (Fig 4A). Similar results were observed for parasite loads by developmental stages of the entire population (Kruskal-Wallis; p = 0.30). However, in *A. aculeata*, N1 and N2 nymphs exhibited significantly lower parasite loads than the other instars, including adults (Kruskal-Wallis; p = 0.02) (Fig 4C). In both *A. butyracea* (p = 0.72) and *M. flexuosa* (p = 0.98), similar parasite loads were found across all

developmental stages (Fig 4B and 4D). When comparing parasite loads of each stage by palm species, N4 captured in *A. aculeata* had a higher parasite load than in the other palm species (Kruskal-Wallis; p = 0.006).

## Feeding sources

Our investigation unveiled that *R. prolixus*, spanning both nymphs and adults, displayed a diverse diet by feeding on various species of arboreal and terrestrial vertebrates, including domestic animals (Fig 5). Arboreal species identified encompassed *Cebus albifrons* (Primates: Cebidae), *Tamandua tetradactyla* (Pilosa: Myrmecophagidae), *Vampyrum spectrum* (Chiroptera: Phyllostomidae), *Didelphis marsupialis*, and *Micoureus demerarae* (Didelphimorphia: Didelphidae). Terrestrial species included *Dasypus novemcinctus* (Cingulata: Dasypodidae) and *Mus musculus* (Rodentia: Muridae). Bugs were also found to feed on humans (*Homo sapiens*; Primates: Hominidae) and domestic animals such as dogs (*Canis lupus familiaris*; Carnivora: Canidae), cats (*Felis catus*; Carnivora: Felidae), donkeys (*Equus asinus*; Perissodactyla: Equidae), cattle (*Bos taurus*; Artiodactyla: Bovidae), and pigs (*Sus scrofa* domestica; Artiodactyla: Suidae).

Most feeding sources were shared among insects captured in all three palm species. However, *Mus musculus* was exclusively found as a blood source for *R. prolixus* adults captured in *A. aculeata* and *A. butyracea*. Additionally, *Cebus albifrons* was only found in nymphs of insects captured in *A. butyracea* (Fig 5). *Tamandua tetradactyla* was identified as a food source in N1 for *A. butyracea* and *M. flexuosa*, and in N3 for *A. aculeata*. Furthermore, horses (*Equus asinus*) were identified as a food source for N2 captured in *A. butyracea*, in all nymphal stages of *M. flexuosa* insects, and in females and N4 of *A. aculeata*.

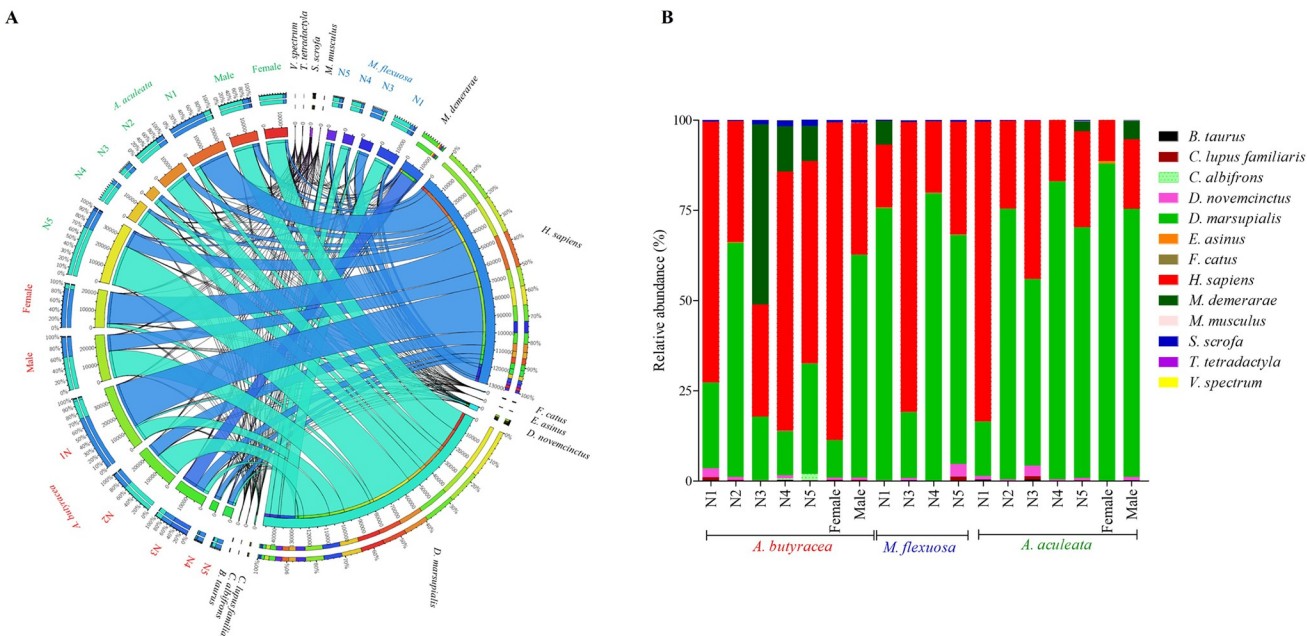

**Fig 5. CIRCOS transmission network.** (A) Shows the food sources found in individuals of *R. prolixus* collected from each palm species and by age structure. The age structure of *R. prolixus* is differentiated by color according to the palm species where it was captured; *A. butyracea* (red), *A. aculeata* (green) and *M. flexuosa* (blue). (B) Relative abundances of food sources of the age structure of *R. prolixus* sampled in each of the palm species. N1-N5, represent the nymphal stages.

*Didelphis marsupialis* emerged as the most abundant and frequently identified feeding source, followed by humans (*H. sapiens*) and *Dasypus novemcinctus*. The least abundant food sources were *Tamandua tetradactyla* and *Vampyrum spectrum* (Fig 5B). The inverse Simpson alpha diversity analysis (D-1) for vertebrates showed no significant differences per palm species (D1 = 0.90–0.92). The Shannon diversity index indicated a low diversity of food sources for the three palm species, with values of H´ = 0.82 for *M. flexuosa*, 1.16 for *A. aculeata*, and 0.82 for *A. butyracea*, without significant differences ($X^2$ = 0.12; p = 0.93).

## Discussion

*Rhodnius prolixus* is well-established as a species closely associated with palm trees, with the palm ecotope providing favorable conditions for its populations. The structural analysis of palms in this study reveals that, while all palm species are susceptible to infestation by *R. prolixus*, *A. butyracea* stands out by offering superior microhabitat conditions for the colonization and establishment of triatomines. Factors such as leaf number and crown volume contribute to *A. butyracea's* suitability for *R. prolixus*. Additionally, the recorded temperature and humidity levels for the three studied palm species suggest that *A. butyracea* provides a more stable climatic environment, contributing to the observed population densities, in contrast to the other assessed palm species. Therefore, variations in habitat availability and microclimatic conditions are proposed as key factors influencing the differing population densities of *R. prolixus*. These findings align with previous observations by Abad-Franch et al. [14], which suggested that palms with higher biomass composition tend to exhibit increased population densities [14,15,22,59–61].

The infestation, colonization, and density indices suggest a preference of *R. prolixus* for *A. butyracea* and *A. aculeata*, showcasing its ability to colonize all three palm species. Notably, *A. butyracea* emerges as a significant ecotope for this triatomine in the Eastern plains, likely due to superior habitat conditions. This association is evident in the presence of all developmental stages and higher insect densities observed (Fig 3B). The presence of all three palm species, especially *A. butyracea*, poses a risk factor as a source of insect population foci for *R. prolixus*, potentially leading to the transmission of *T. cruzi* to inhabitants in these palm-rich environments. This risk is accentuated by the consistently high population densities of the vector across all examined palm species. Importantly, these findings align with earlier studies in the Colombian Orinoco [23,37,38,41,62] and Brazilian Amazonia, highlighting an established association between *A. butyracea* and *R. prolixus* [15,17,27].

Variations in developmental stages among palm species may reflect differences in food availability, intricately tied to habitat accessibility within palm crowns [14,31]. Our investigation uncovered lower microhabitat availability in *M. flexuosa* and reduced microclimatic stability in *A. aculeata*, which could potentially clarify the observed population behavior and developmental stages. Although *R. prolixus* has demonstrated the capability to infest all three palm species, it tends to achieve larger populations in *A. butyracea*. These findings align with similar studies conducted in the Colombian and Venezuelan plains [38,41,59,61,63], consistently revealing higher population densities of *R. prolixus* in *A. butyracea*. This underscores the significance of *A. butyracea* as a critical habitat for *R. prolixus*, emphasizing its importance as a resident of this specific palm species.

Concerning *T. cruzi* infection, *R. prolixus* collected in *M. flexuosa* exhibited a slightly lower percentage, while *A. butyracea* hosted insects with the maximum parasite infection rate. This aligns with previous reports for the region by Urbano et al. (85.2%) [23], Calderón et al. (28%) [34], Rendón et al. (60.2%) [37], and Velásquez-Ortiz et al. (84%) [52]. The presence of these

wild palms in the zone [29] poses a direct threat to *T. cruzi* transmission in rural areas. This is exacerbated by most human settlements being established on fragmented patches resulting from palm felling in the peridomicile, facilitating triatomine intrusion and increasing the likelihood of parasite transmission [38,64–66]. Furthermore, the identification of all positive samples as TcI indicates continuous circulation of the parasite in all three palm species, underscoring their role as ecotopes for infected insects. Despite population-level parasite burden values and developmental stages being higher than previously reported in the zone [52], no significant differences were noted.

Regarding feeding sources, there is evidence that ecosystem transformations lead to changes in the composition of animal communities in a given area [35]. In human-modified landscapes, few species prove resilient to environmental changes [67]. For vertebrates associated with forests, the alteration of their habitats directly impacts species composition, given that most are specialists [31]. This could elucidate the observed low vertebrate composition as feeding sources for the insects captured in the three palm species. Our results diverge from those reported by Arias-Giraldo et al. [33] and Calderón et al. [34], where the diversity of triatomine food sources was higher. However, the fact that insects captured in the three palm species, at all developmental stages, shared their feeding sources suggests that the vector could establish itself in any of the wild palms studied. Additionally, food availability tends to be lower in transformed ecosystems, considering fauna migration as a consequence of anthropogenic action [67]. Consequently, insects established in areas under anthropogenic interference consistently face limiting factors in food supply [68,69]. Under starvation conditions, they may extend their range towards locations such as human dwellings and peridomiciliary environments where they can satisfy their needs. This could explain why the main sources were humans and domestic animals.

Our results align with other studies conducted in the Orinoco and Amazonia [33–35,37]. However, it is crucial to highlight that the sampling was conducted in palms far from houses. In this context, it has been determined that distances greater than 110 m are hardly viable for the directional flight of vectors such as *R. prolixus* [70]. Therefore, the question of how nymphal stages, especially N1-N3, approach humans and domestic animals for feeding remains unanswered. However, various studies have identified human blood in wild nymphs of other *Rhodnius* species, such as *R. pallescens* in Panama [71] and Colombia [72], *R. ecuadoriensis* in Ecuador [73], and *R. stali* and *R. montenegrensis* in Bolivia [74]. These findings suggest that feeding behaviors, like kleptohematophagy, could shed light on the presence of food sources from domestic environments in nymphal stages with limited mobility [70]. Additionally, the intriguing relationship between *R. prolixus* and humans in Casanare deserves attention. The coexistence of *R. prolixus* with other species, known for their tendency to feed on human blood, is particularly noteworthy [72,75]. This phenomenon may be explained by human activities, such as agriculture, at the collection sites. Palms located near homes create favorable conditions for vectors to come into contact with human hosts. Principio del formulario.

Given the high sensitivity of Next-Generation Sequencing (NGS) techniques and prior reports of human DNA contaminants in mosquitoes [76], we acknowledge the growing significance of addressing potential sources of contamination. Throughout our entire procedure, we have implemented stringent controls to forestall contamination, ensuring the reliability of our results. To mitigate this risk, we adhere to rigorous laboratory protocols aligned with those employed in the molecular diagnosis of infectious agents, coupled with the implementation of comprehensive sample handling procedures. In addition, researchers involved in insect processing and handling use all protective barriers, such as gowns, gloves, caps, and masks. Additionally, samples, prior to extraction, are manipulated in a type A2 biosafety cabinet and

subsequently extracted by an automated extraction system. In our study, we implemented rigorous measures, including the use of 5 controls during DNA extraction (which automated for decrease the risk of contamination) and 10 controls (Eukaryotic DNA-free water) during PCR for the assessment of food sources (12S). Furthermore, we incorporated 10 controls from colony-fed individuals with *Gallus gallus*, where no traces of human DNA were detected, ensuring the likely absence of contamination in the laboratory. Nevertheless, it is crucial to acknowledge that the identification of human-origin food sources in nymphal stages with limited mobility could potentially result from contamination during field collection. Therefore, the incorporation of negative controls during the collection of triatomines in future studies will be essential [76].

On the other hand, species like *D. marsupialis* have consistently been reported as triatomine feeding sources [27,40], corroborating our own findings. However, the noteworthy revelation that humans rank as the second most abundant and frequent source underscores the imperative to delve deeper into the mechanisms by which these insects, located in extra-peridomiciliary palms and feed on humans. Furthermore, the inclusion of species such as *D. novemcinctus* as the third most abundant implies that adults and other life stages may engage in migration to the basal layers of the ecosystem. This behavior suggests a predominant feeding pattern on terrestrial mammals, with sporadic feeding on species like *V. spectrum*, which is exclusively arboreal.

The population density of *R. prolixus* observed in *A. butyracea* exceeded that found in *A. aculeata* and *M. flexuosa*. Additionally, the representativity of developmental stages varied depending on the sampled palm, with lower insect densities recorded in *M. flexuosa*. This pattern may be linked to the structural physiognomy of the palms, as *A. butyracea* likely offers a more favorable microhabitat and microclimatic stability compared to the other two palm species. Remarkably, the insects collected exhibited almost identical blood sources, suggesting a lack of differentiation in the population diet within the study zone. The diets encompassed animals such as *C. albifrons*, *D. novemcinctus*, *V. spectrum*, *E. asinus*, *C. lupus familiaris*, and humans.

Our findings, coupled with the elevated values of entomological indicators, underscore a substantial risk of Chagas disease transmission through human interactions with triatomine ecotopes. These ecotopes result from transformation processes in the natural ecosystem. Despite the inherently wild nature of these triatomine species, the frequent utilization of their leaves in home construction and their fruits as feed for domestic animals heightens the risk of parasite transmission to humans. However, this risk could be ameliorated if *M. flexuosa* is favored against *A. butyracea*. Increasing the density of the palm tree with less bug infestation rate and density and bug parasite prevalence could lead to a reduction in vector density and *T. cruzi* transmission. Since *M. flexuosa* is an important microhabitat for several wildlife species, this measure could also provide a conservation value for the region.

## Conclusion

This study revealed significant variations in the microclimate and physiognomy of three palm species. While *A. butyracea* exhibited higher DBH values and microclimate stability, *A. aculeata* and *M. flexuosa* displayed distinct characteristics. *Rhodnius prolixus* demonstrated higher densities in *A. butyracea*, impacting entomological indices and *T. cruzi* infection rates. The feeding sources analysis unveiled diverse vertebrates, emphasizing the relevance of these palms in the ecology of *T. cruzi* vectors. These findings contribute valuable insights into the complex interactions between palm characteristics and the epidemiology of Chagas disease. Furthermore, they suggest the presence of species such as *A. butyracea* and *A. aculeata* as a risk factor

for the transmission of *T. cruzi* to people who settle in rural areas where the plant composition of the landscapes is dominated by these species of palms. For this reason, it is urgent to design strategies to control the intrusion of triatomines into homes in these anthropogenic areas.

## Supporting information

**S1 Table. List of pools selected for further molecular analysis.**
(XLSX)

**S2 Table. Statistical analysis comparing palm species and *R. prolixus* developmental stages.**
(XLSX)

## Author Contributions

**Conceptualization:** Plutarco Urbano, Camila González, Juan David Ramírez.

**Data curation:** Plutarco Urbano, Laura Vega, Marina Muñoz.

**Formal analysis:** Plutarco Urbano, Carolina Hernández, Natalia Velásquez-Ortiz, Nathalia Ballesteros, Luisa Páez-Triana, Laura Vega, Vanessa Urrea, Angie Ramírez, Marina Muñoz, Carlos N. Ibarra-Cerdeña, Camila González, Juan David Ramírez.

**Funding acquisition:** Plutarco Urbano, Marina Muñoz, Juan David Ramírez.

**Investigation:** Plutarco Urbano, Carolina Hernández, Natalia Velásquez-Ortiz, Nathalia Ballesteros, Luisa Páez-Triana, Laura Vega, Vanessa Urrea, Angie Ramírez, Marina Muñoz, Carlos N. Ibarra-Cerdeña, Camila González.

**Methodology:** Plutarco Urbano, Carolina Hernández, Natalia Velásquez-Ortiz, Nathalia Ballesteros, Luisa Páez-Triana, Laura Vega, Vanessa Urrea, Angie Ramírez, Marina Muñoz, Carlos N. Ibarra-Cerdeña, Camila González, Juan David Ramírez.

**Resources:** Plutarco Urbano, Carolina Hernández, Natalia Velásquez-Ortiz, Nathalia Ballesteros, Luisa Páez-Triana, Vanessa Urrea, Marina Muñoz, Carlos N. Ibarra-Cerdeña, Camila González, Juan David Ramírez.

**Supervision:** Carolina Hernández, Marina Muñoz, Carlos N. Ibarra-Cerdeña, Camila González, Juan David Ramírez.

**Validation:** Plutarco Urbano, Marina Muñoz.

**Visualization:** Plutarco Urbano.

**Writing – original draft:** Plutarco Urbano.

**Writing – review & editing:** Carolina Hernández, Natalia Velásquez-Ortiz, Nathalia Ballesteros, Luisa Páez-Triana, Laura Vega, Vanessa Urrea, Angie Ramírez, Marina Muñoz, Carlos N. Ibarra-Cerdeña, Camila González, Juan David Ramírez.

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
