## [Decision Letter · Decision Letter 0]

27 Nov 2023

Dear Dr. Ramírez,

Thank you very much for submitting your manuscript "Variation in the transmission ecology of Trypanosoma cruzi in Rhodnius prolixus (Reduviidae: Triatominae) among palm-tree species in the Colombian Orinoco" for consideration at PLOS Neglected Tropical Diseases. As with all papers reviewed by the journal, your manuscript was reviewed by members of the editorial board and by several independent reviewers. In light of the reviews (below this email), we would like to invite the resubmission of a significantly-revised version that takes into account the reviewers' comments. 

The manuscript was evaluated by three specialists in the area which acknowledged the importance of the paper. However, all the reviewers raised concerns about the English. Therefore, the authors must submit the manuscript to a professional English reviewer before resubmission. In addition, another important concern is the possibility of contamination of NSG samples, raised by reviewers #2 and #3. Please, address this concern carefully and state any limitations that may impact the interpretation of results. Address other suggestion and concerns in a point-by-point response.

We cannot make any decision about publication until we have seen the revised manuscript and your response to the reviewers' comments. Your revised manuscript is also likely to be sent to reviewers for further evaluation.

Sincerely,

Helton C. Santiago, M.D., Ph.D

Academic Editor

Esther Schnettler

Section Editor

The manuscript was evaluated by three specialists in the area which acknowledged the importance of the paper. However, all the reviewers raised concerns about the English. Therefore, the authors must submit the manuscript to a professional English reviewer before resubmission. In addition, another important concern is the possibility of contamination of NSG samples, raised by reviewers #2 and #3. Please, address this concern carefully and state any limitations that may impact the interpretation of results. Address other suggestion and concerns in a point-by-point response.

Reviewer's Responses to Questions

**Key Review Criteria Required for Acceptance?**

**Methods**

-Are the objectives of the study clearly articulated with a clear testable hypothesis stated?

-Is the study design appropriate to address the stated objectives?

-Is the population clearly described and appropriate for the hypothesis being tested?

-Is the sample size sufficient to ensure adequate power to address the hypothesis being tested?

-Were correct statistical analysis used to support conclusions?

-Are there concerns about ethical or regulatory requirements being met?

Reviewer #1: The hypothesis is not stated in the manuscript;

The materials and methods are in accordance with the objectives and obtained results;

The samples are robust and the obtained results where statistically analyzed giving support to the conclusions;

There are no concerns about ethical requirements.

Reviewer #2: -Are the objectives of the study clearly articulated with a clear testable hypothesis stated? yes, they are

-Is the study design appropriate to address the stated objectives? yes, it is

-Is the population clearly described and appropriate for the hypothesis being tested? yes, it is

-Is the sample size sufficient to ensure adequate power to address the hypothesis being tested? see comments

-Were correct statistical analysis used to support conclusions? yes, they were

-Are there concerns about ethical or regulatory requirements being met? As far as I understand, I have no concern in this sense (I do not belong to their country)

Reviewer #3: Please describe methods taken to ensure there was no contamination during extraction and PCR - e.g. were negative controls used, were assays run in duplicate/triplicate, were results confirmed with second molecular target, etc. Additionally, what measures were taken during bug processing/extraction to prevent contamination with human DNA that would be detected during vertebrate host analysis? 

Some additional specific comments:

Line 151-142: It would be helpful here to briefly elaborate on the entomological indices, infestation, clustering, colonization, and density so the reader doesn't have to look up the citation just to follow your results. For example, later in the manuscript you describe density index as "number of individuals per sampled palm" - this type of detail could be included here perhaps in parentheses for all of the different indices.

Line 180: Please clarify if bugs selected for feeding sources analysis were only the T.cruzi-positive bugs? Why? And specify whether each of the pools were formed from bugs all from the same tree, or from multiple trees of the same species. 

Line 193: State what statistical program was used for these analyses - later you mention R but not until specifically talking about creating the graphs.

**Results**

-Does the analysis presented match the analysis plan?

-Are the results clearly and completely presented?

-Are the figures (Tables, Images) of sufficient quality for clarity?

Reviewer #1: The results are clear, well presented and the statistical analysis gives support to the conclusion

The tables and figs are clear and in good quality

Reviewer #2: I have only concerns about the way molecular material was processed.

Reviewer #3: Very nice data visualization! It may be helpful to add the Orinoco region to the map (Fig 1), as this region is mentioned several times but it was unclear to this reader exactly how that region relates to the Casanare district.

**Conclusions**

-Are the conclusions supported by the data presented?

-Are the limitations of analysis clearly described?

-Do the authors discuss how these data can be helpful to advance our understanding of the topic under study?

-Is public health relevance addressed?

Reviewer #1: There is no specific topic for the conclusion. We recommend a specific topic listing the main results.

The discussion is well contextualized based in the literature 

A few suggestions were listed in the comments bellow for the conclusions

Reviewer #2: My primary concern pertains to the potential for the conclusions to inadequately represent the scientific reality.

Reviewer #3: Please add some discussion about the possibility of contamination and whether you think this is likely or unlikely to explain the high frequency of apparent human blood meals.

Please add the details of the prevalences previously reported among R. prolixus collected from palm trees in this zone, this is needed to see how your results compare (line 352).

**Editorial and Data Presentation Modifications?**

Reviewer #1: Please check comments bellow

Reviewer #2: The text must be dramatically improved. See below: 

Ln 322: higher [..insect] densities.

Ln 333-334: “that the [the presence of all] three palm species studied”.

Ln 334: “…as source of insect population foci of R. prolixus to transmit CD to people living …” [rewrite]

Ln 334: delete “larger”.

Ln 337-338: replace “host” by “inhabitant of A. butyracea”. The word “host” is often used for food sources.

Ln 357-358: The verb "represent" agrees with which subject?

Ln 361-363: I do not see a connection between the parasite genotype and the epidemiological importance.

Ln 377: Delete the “the”

Ln 378: “perturbation” can be replaced by “anthropogenic action”.

Ln 379: “perturbed” = areas under anthropogenic interference".

Ln 381: “This could explain that the main” = “This could explain why the main”.

Ln 392: delete “about”.

Ln 397: replace “those” by “that”.

Ln 399: “lower [insect] densities”.

Ln 40-401: A. butyracea must be in italics.

Ln 406: “indices” may be replaced by “indicators”.

Ln 407: delete the “the” before “humans”.

Reviewer #3: Line 37-38: This sentence doesn't make sense as written - Insects... had higher abundances of R. prolixus.

Line 41-42: The us of both "all examined insects" and "R. prolixus" in this sentence makes it sound like multiple species of insects were studied, but I don't think that is correct.

Line 78: I'm not sure what is meant by "frequent reproductive events and leaves abscission", please clarify, especially whether reproductive events refers to the plant or the bugs?

Line 86-90: This sentence is confusing, please revise for clarity.

Line 291: I believe Equus asinus would more commonly be referred to as donkeys, not horses (Equus caballus).

Line 333-334: I don't think that the term 'significant risk factor' is appropriate here, as you have not conducted an epidemiological study looking at transmission in humans relative to the presence of palms. Could be revised as "potentially pose a significant risk for the transmission".

Additionally, the manuscript would benefit from some additional editing for grammar.

**Summary and General Comments**

Reviewer #1: Review Plos NTD

Manuscript number: PNTD-D-23-00601

General Comments:

The manuscript entitled “Variation in the transmission ecology of Trypanosoma cruzi in Rhodnius prolixus (Reduviidae: Triatominae) among palm-tree species in the Colombian Orinoco” is well presented, written, discussed. The M & M are in accordance with the objectives and obtained results. The English language is nice however, a few mistakes could be found in the text therefore, a review is suggested. In the introduction a few references are missing. There is no contextualization neither on the triatomine vectors nor in the T. cruzi. Minor comments are listed bellow. Despite the fact the association of R. prolixus with distinct palm species has been studied by several authors, this manuscript presents relevant and original information and therefore, it has merits to be recommended for publication in the PNTD

Specific comments: 

Title: I would suggest changing it for: “ Transmission ecology of Trypanosoma cruzi in (or by?) Rhodnius prolixus (Reduviidae: Triatominae) infesting palm-tree species in the Colombian Orinoco, indicates risks to human populations”

Background- We suggest writing the scientific names in full, because it is the first time of the species citation

Author summary- Please revise the verb in the sentence “Remarkably, it was found that the 56 kissing bugs studied, feeds on wild and domestics animal…” I think the correct is “feed”

Introduction- In the first paragraph, the authors briefly contextualize the Chagas disease however, very little is mentioned presenting either the etiologic agent or the triatomine vectors. We recommend including some general comments and references for both: 

A review of the taxonomy and biology of Triatominae subspecies (Hemiptera: Reduviidae) – Parasitology Research

Do the new triatomine species pose new challenges or strategies for monitoring Chagas disease? An overview from 1979-2021 – Memórias do Instituto Oswaldo Cruz

Evolution, Systematics, and Biogeography of the Triatominae, Vectors of Chagas Disease - Chapter

Ecology and diversity of Trypanosoma cruzi – Acta Tropica

Trypanosoma Cruzi: An Ancient and Successful Enzootic Parasite - Infectious Tropical Diseases and One Health in Latin America

Also, after contextualizing the importance of the palms for the distribution, ecology and so on we suggest including some words on the importance of R, prolixus. This would improve the introducrion

Materials and methods

I would suggest including the total range of the studied area and how the humans occupy the region, also the approximate number of humans living in the area, and the number of domiciliary unities. This kind of information is highly relevant for the study carried out.

I would suggest “Triatomine sampling” instead of “Triatominae sampling”. In this topic it would be important to mention the season on which the captures were carried out. The dry and the wet seasons have relevant influences on the field work results.

Results

Line 272- “development stages” please change for “developmental stages”

Line 278- In the sentence “…than in the other palm species” I think the preposition “in” should be crossed out

Line 280- “sampled processed” or “samples processed”?

Line 314- “Represents the nymph ages” please change for “represent the nymphal stages”

Discussion

Line 343- In the sentence “Therefore, although R. prolixus is able to infest all three palm 344 species, in A. butyracea they reach bigger larger populations…” I would suggest “Therefore, although R. prolixus is able to infest all three palm species, in A. butyracea it reaches larger populations”

Line 347- In the sentence “R. prolixus achieves large population densities” please consider changing “R. prolixus achieves higher population densities “

Line 281- “peridomicile environments “ please change for “ peridomiciliary environments”

Conclusion 

I would suggest including a conclusion topic. The last paragraph of the discussion could be extended listing and highlighting the most relevant findings, for instance: the numerous feeding sources including human blood as one of the most important; the obtained results in a silvatic area near human dwellings distinct from previous studies; and the importance of educational programs for people living in the area with high risk of transmission, among other relevant results

Legends

 Suggestion for changes: Table 1- It would be important including the vector species name

Reviewer #2: Major concerns

The study design is commendable as the author conducted a comprehensive R. prolixius sampling of each of the three species of palm trees. Abiotic data has been intricately linked with biotic factors of eco-epidemiological significance. Molecular and statistical techniques were effectively employed in R. prolixius molecular ecology. While I appreciate the study and believe its results are worthy of publication, I do have a significant concern about the unexpectedly high frequency of first to third-stage nymph blood feedings on terrestrial animals - particularly humans. Although the recent NGS-based techniques have enhanced resolution in current studies, their heightened sensitivity increases susceptibility to contaminations. Various authors had already warned about the potential for contamination at any stage of the study involving NGS, from the field to the laboratory (see https://doi.org/10.1371/journal.pntd.0004512). I believe that this issue should not be treated as a taboo. I would consider excluding human feedings. This could be done by simply adding a statement in the Materials and Methods section, indicating that “Unexpected feedings, such as those in humans at nymphal developmental stages, were excluded from the analyses.”. Unfortunately, this would require rerunning all the analyses, but this would present a more realistic perspective. The authors are required to undertake a thorough revision of the manuscript, with a particular focus on the abstract and discussion sections. In the latter, issues were identified in nearly every line. Despite these criticisms, I reiterate that this study provides valuable information within the scope of PlosNTD.

Minor comments or concerns

Title:

Although frequently used in other studies, the term “transmission ecology” does not reflect the content of the study. 

When first mentioning the genera of palm trees, “Attalea”, “Mauritia”, and “Acrocomia” in the Abstract (ln 28 on), please use the full name, and then provide the abbreviation for brevity.

Ln. 29 M&M: “resulting in 105 samples” of?

Ln 32-34: “ …analyzed using qPCR for T. cruzi detection and discrete typing units' identification. The 33 feeding preferences were determined by sequencing the 12S rRNA gene amplicon using 34 next-generation sequencing.”. This part belongs to the M&M”.

Ln.112-115: The objective statement appears to be rather vague.

Ln 235-246 “Population density of R. prolixus per palm species”: The initial number of 676 R. prolixus individuals collected may seem substantial; however, when these data are stratified by palm species and insect instar, they may prove to be insufficient for robust comparisons. On the other hand, the chosen statistical tests are suitable for handling non-normally distributed data. To enhance clarity, I suggest placing the stratification details in a supplementary file (consider insertirng X, SD, Max., Min) and discussing this in the “Discussion” section.

Ln 262-278: “Infection index and quantification of T. cruzi”: the same for “Population density of R. prolixus per palm species”.

Ln 290: there is double parenthesis “((“.

Discussion

Although it's a matter of writing style, I recommend avoiding referencing figures and tables in the discussion. Thus, the discussion flows more smoothly when referring to the literature.

Ln 406-413: Are any of these palm trees planted by humans? This discussion only makes sense if any of the three species are used for human activities of interest. This paragraph would make much more sense in the context of anthropogenic actions if any one of them is favored by the residents.

Reviewer #3: This is a very nice and interesting paper reporting systematic sampling of three species of palm trees for Rhodnius prolixus, an important vector of Trypanosoma cruzi, with parasite detection and vertebrate host meal analysis. It is certainly of value to the T. cruzi transmission ecology literature. My major concern is that the T.cruzi infection prevalence and human bloodmeal prevalence are very high, so there should be some more details and discussion about how the authors ensured this was not the result of contamination.

PLOS authors have the option to publish the peer review history of their article (what does this mean?). If published, this will include your full peer review and any attached files.

Reviewer #1: No

Reviewer #2: Yes: CEA

Reviewer #3: No
---

## [Decision Letter · Decision Letter 1]

19 Jan 2024

Dear Dr. Ramírez,

Thank you very much for submitting your manuscript "Transmission ecology of Trypanosoma cruzi by Rhodnius prolixus (Reduviidae: Triatominae) infesting palm-tree species in the Colombian Orinoco, indicates risks to human populations" for consideration at PLOS Neglected Tropical Diseases. As with all papers reviewed by the journal, your manuscript was reviewed by members of the editorial board and by several independent reviewers. The reviewers appreciated the attention to an important topic. Based on the reviews, we are likely to accept this manuscript for publication, providing that you modify the manuscript according to the review recommendations. 

The improvement and importance of the manuscript were acknowledged by all the reviewers. However, all the reviewers still requested minor revision. While few punctuation suggestions were made, reviewers 2 and 3 still show concerns about sample contamination, especially during field capture. The authors should at least mention such possibility as a limitation with the opportunity to strengthen the reasons why the contamination is unlikely.

Sincerely,

Helton C. Santiago, M.D., Ph.D

Academic Editor

Esther Schnettler

Section Editor

The improvement and importance of the manuscript were acknowledged by all the reviewers. However, all the reviewers still requested minor revision. While few punctuation suggestions were made, reviewers 2 and 3 still show concerns about sample contamination, especially during field capture. The authors should at least mention such possibility as a limitation with the opportunity to strengthen the reasons why the contamination is unlikely.

Reviewer's Responses to Questions

**Key Review Criteria Required for Acceptance?**

**Methods**

-Are the objectives of the study clearly articulated with a clear testable hypothesis stated?

-Is the study design appropriate to address the stated objectives?

-Is the population clearly described and appropriate for the hypothesis being tested?

-Is the sample size sufficient to ensure adequate power to address the hypothesis being tested?

-Were correct statistical analysis used to support conclusions?

-Are there concerns about ethical or regulatory requirements being met?

Reviewer #1: The manuscript was greatly improved. Minor issues were detected and must be corrected;

THe objectives of the study are clearly articulated with a clear testable hypothesis stated;

There are no concerns about ethical requirements;

Population is clearly described and the sample size is sufficient;

Reviewer #2: The authors presented compelling arguments to refute the possibility of contamination in the laboratory. However, they failed to acknowledge the information I provided (See https: Logue et al. 2016), indicating that contamination could also occur in the field, during insect captures. This poses a significant challenge, particularly in studies involving highly sensitive methods to detect food sources. This potential source of contamination cannot be dismissed and warrants further discussion. This is especially crucial since nymphs in their early developmental stages exhibit extremely limited mobility. This is challenging to conceive that N1 can descend from a palm tree to feed on humans and then return to the tree - considering the observed high prevalence of feeding on humans during these sessile stages. In the discussion, it is important to acknowledge and consider the possibility of contamination in the field, as highlighted, to ensure an accurate evaluation of the research findings.

See: Logue K, Keven JB, Cannon MV, Reimer L, Siba P, Walker ED, et al. (2016) Unbiased Characterization of Anopheles Mosquito Blood Meals by Targeted High-Throughput Sequencing. PLoS Negl Trop Dis 10(3): e0004512. https://doi.org/10.1371/journal.pntd.0004512.

Reviewer #3: The additions to the methods to describe positive and negative controls and the additional information provided to reviewers on the separate stations for processing are encouraging. However, I would still like to see some details on the precautions taken to prevent human contamination of samples prior to extraction - what PPE was used (gloves, sleeves, lab coat, hair net?), were extractions conducted in biosafety cabinet or benchtop, etc.

How were the 115 triatomines subjected to DNA extraction selected from the 676 total collected? Was this a systematic process? Random? Stratified? It does not appear that the number of bugs tested from each tree species was representative of the overall distribution of number of bugs collected. Was a power calculation or any other stats done to determine the sample size that would be tested?

**Results**

-Does the analysis presented match the analysis plan?

-Are the results clearly and completely presented?

-Are the figures (Tables, Images) of sufficient quality for clarity?

Reviewer #1: M&M are in accordance with the results and tables and figs are ok.

Reviewer #2: The authors present the results they have obtained, but it's important to acknowledge that these findings may be biased.

Reviewer #3: (No Response)

**Conclusions**

-Are the conclusions supported by the data presented?

-Are the limitations of analysis clearly described?

-Do the authors discuss how these data can be helpful to advance our understanding of the topic under study?

-Is public health relevance addressed?

Reviewer #1: The discussion is pertinent

Conclusions were clearly included

Reviewer #2: Logue et al. (2016) mentioned "Here, we included 30 extraction (water) controls that were all negative suggesting very low levels of laboratory contamination (if any). An interesting complementary control, which would also control for field contamination, would be to analyze male mosquitoes collected at the same time."

For mosquitoes, only females feed on humans. The authors could insert a similar statment, such as: 

 Here, we incorporated [...insert your cautions...] controls, all of which yielded negative results, indicating minimal to no laboratory contamination. A complementary control, which could additionally address potential field contamination, would involve analyzing non-hematophagous insects collected simultaneously [modify for your convenience]

Reviewer #3: It's not clear that there is good statistical support for the statement in lines 395-396 and 398-401 regarding differences in triatomine infection prevalence between tree species given the sample size tested was a subset of the total insects collected and details are missing on how that subset was selected.

**Editorial and Data Presentation Modifications?**

Reviewer #1: Background

A coma is missing… Chagas disease, affecting approximately eight million individuals in tropical regions, is primarily transmitted by vectors.

I would suggest changing “resides” by inhabits or infests or colonizes

Conclusion 

The term “individuals” is a bit vague. I would suggest using a more precise term

High population densities and infection rates were observed in each examined palm tree species. I would suggest including: High population densities and infection rates of R. prolixus were observed in each examined palm tree species.

M&M

In the line 170 the species name is written in full. Please change it to R. prolixus and revise all manuscript.

Discussion

Line 363- Please correct Attalea butyracea “A. butyracea”

Line 477- Please correct R. prolixus ; after a period or to initiate a paragraph please write the scientific name in full.

Reviewer #2: The manuscript is well presented

Reviewer #3: (No Response)

**Summary and General Comments**

Reviewer #1: (No Response)

Reviewer #2: See the comments for Methods

Reviewer #3: The manuscript has been improved by the revisions conducted. There are just a few additional issues needing clarification.

PLOS authors have the option to publish the peer review history of their article (what does this mean?). If published, this will include your full peer review and any attached files.

Reviewer #1: No

Reviewer #2: Yes: CEA

Reviewer #3: No

Figure Files:

Data Requirements:

Reproducibility:

References

---

## [Decision Letter · Decision Letter 2]

8 Feb 2024

Dear Dr. Ramírez,

We are pleased to inform you that your manuscript 'Transmission ecology of Trypanosoma cruzi by Rhodnius prolixus (Reduviidae: Triatominae) infesting palm-tree species in the Colombian Orinoco, indicates risks to human populations' has been provisionally accepted for publication in PLOS Neglected Tropical Diseases.

Best regards,

Helton C. Santiago, M.D., Ph.D

Academic Editor

Esther Schnettler

Section Editor

Reviewer #1 still made few suggestions, which we consider may be addressed by the authors, if they consider appropriate, during the production phase.

Reviewer's Responses to Questions

**Key Review Criteria Required for Acceptance?**

**Methods**

-Are the objectives of the study clearly articulated with a clear testable hypothesis stated?

-Is the study design appropriate to address the stated objectives?

-Is the population clearly described and appropriate for the hypothesis being tested?

-Is the sample size sufficient to ensure adequate power to address the hypothesis being tested?

-Were correct statistical analysis used to support conclusions?

-Are there concerns about ethical or regulatory requirements being met?

Reviewer #1: Objectives, methodology, results, discussion and conclusion are in accordance wiht the hypothesis being tested

Reviewer #2: In this version, the authors acknowledge the possibility of field contamination. I support the publication of this manuscript.

Reviewer #3: (No Response)

**Results**

-Does the analysis presented match the analysis plan?

-Are the results clearly and completely presented?

-Are the figures (Tables, Images) of sufficient quality for clarity?

Reviewer #1: Results are well presented and discussed

Reviewer #2: yes

Reviewer #3: (No Response)

**Conclusions**

-Are the conclusions supported by the data presented?

-Are the limitations of analysis clearly described?

-Do the authors discuss how these data can be helpful to advance our understanding of the topic under study?

-Is public health relevance addressed?

Reviewer #1: Conclusions are in accordance with the results

Reviewer #2: Yes, they are.

Reviewer #3: (No Response)

**Editorial and Data Presentation Modifications?**

Reviewer #1: Minor revision

Reviewer #2: needless

Reviewer #3: (No Response)

**Summary and General Comments**

Reviewer #1: The manuscript was greatly improved.

Unfortunately, I did not detect in the very first round the manuscript is not related to Chagas disease (Epidemiology, symptoms, mortality, and morbidity) but to the T. cruzi transmission ecology. Sorry for that!

Chagas disease is a very complex disease in several aspects therefore, in order to get the manuscript more technically precise, I suggest replacing the several expressions like “Chagas disease vectors” Chagas disease transmission” “risk of Chagas disease” by “T. cruzi vectors”, “T. cruzi transmission”, “risk of T. cruzi…” Please check all manuscript

Examples:

Author summary- several sentences to be modified mainly the first and last ones

Introduction line 6;

page 3 line3; page 4 line 1;

page 10 line 40; page 12 line 487; page 13 line 502

Results- page 9- Feeding sources: In order to be more clear please number the total species functioning as food sources (13 species in total?)

Reviewer #2: In this version, the authors acknowledge the possibility of field contamination. I support the publication of this manuscript.

Reviewer #3: My previous comments have been satisfactorily addressed and I have no further comments.

PLOS authors have the option to publish the peer review history of their article (what does this mean?). If published, this will include your full peer review and any attached files.

Reviewer #1: No

Reviewer #2: **Yes: **CEA

Reviewer #3: No

---

## [Editor Report · Acceptance letter]

13 Feb 2024

Dear Dr. Ramírez,

We are delighted to inform you that your manuscript, "Transmission ecology of Trypanosoma cruzi by Rhodnius prolixus (Reduviidae: Triatominae) infesting palm-tree species in the Colombian Orinoco, indicates risks to human populations," has been formally accepted for publication in PLOS Neglected Tropical Diseases.

Best regards,

Shaden Kamhawi

co-Editor-in-Chief

Paul Brindley

co-Editor-in-Chief
